# Cold Atmospheric Plasma Increases Temozolomide Sensitivity of Three-Dimensional Glioblastoma Spheroids via Oxidative Stress-Mediated DNA Damage

**DOI:** 10.3390/cancers13081780

**Published:** 2021-04-08

**Authors:** Priyanka Shaw, Naresh Kumar, Angela Privat-Maldonado, Evelien Smits, Annemie Bogaerts

**Affiliations:** 1Research Group PLASMANT, Department of Chemistry, University of Antwerp, 2610 Antwerp, Belgium; nash.bms@gmail.com (N.K.); Angela.PrivatMaldonado@uantwerpen.be (A.P.-M.); 2Solid Tumor Immunology Group, Center for Oncological Research (CORE), Integrated Personalized and Precision Oncology Network (IPPON), University of Antwerp, 2610 Antwerp, Belgium; Evelien.Smits@uza.be; 3National Institute of Pharmaceutical Education and Research, Guwahati, Guwahati 781125, Assam, India

**Keywords:** temozolomide, cold atmospheric plasma, glioblastoma, reactive oxygen and nitrogen species, oxidative stress, spheroid model, GSH/GPX4, DNA damage

## Abstract

**Simple Summary:**

Cold atmospheric plasma (CAP) is gaining increasing interest for cancer treatment, for a wide range of cancer types. The studies performed with CAP as a standalone treatment modality serve as evidence that it can also be a suitable candidate for combination therapy. Temozolomide (TMZ) is used as the gold standard drug for glioblastoma treatment, one of the most aggressive malignant brain tumors in adults that remains incurable despite treatment advances. In this study, we explore whether CAP, a cocktail of reactive oxygen and nitrogen species, can amplify the cytotoxic effect on both TMZ-sensitive and TMZ-resistant glioblastoma multiforme (GBM) in three-dimensional tumor-like tissues through inhibiting the glutathione (GSH)/ glutathione peroxidase 4 (GPX4) antioxidant machinery, which can further lead to DNA damage.

**Abstract:**

Glioblastoma multiforme (GBM) is the most frequent and aggressive primary malignant brain tumor in adults. Current standard radiotherapy and adjuvant chemotherapy with the alkylating agent temozolomide (TMZ) yield poor clinical outcome. This is due to the stem-like properties of tumor cells and genetic abnormalities in GBM, which contribute to resistance to TMZ and progression. In this study, we used cold atmospheric plasma (CAP) to enhance the sensitivity to TMZ through inhibition of antioxidant signaling (linked to TMZ resistance). We demonstrate that CAP indeed enhances the cytotoxicity of TMZ by targeting the antioxidant specific glutathione (GSH)/glutathione peroxidase 4 (GPX4) signaling. We optimized the threshold concentration of TMZ on five different GBM cell lines (U251, LN18, LN229, U87-MG and T98G). We combined TMZ with CAP and tested it on both TMZ-sensitive (U251, LN18 and LN229) and TMZ-resistant (U87-MG and T98G) cell lines using two-dimensional cell cultures. Subsequently, we used a three-dimensional spheroid model for the U251 (TMZ-sensitive) and U87-MG and T98G (TMZ-resistant) cells. The sensitivity of TMZ was enhanced, i.e., higher cytotoxicity and spheroid shrinkage was obtained when TMZ and CAP were administered together. We attribute the anticancer properties to the release of intracellular reactive oxygen species, through inhibiting the GSH/GPX4 antioxidant machinery, which can lead to DNA damage. Overall, our findings suggest that the combination of CAP with TMZ is a promising combination therapy to enhance the efficacy of TMZ towards the treatment of GBM spheroids.

## 1. Introduction

Glioblastoma multiforme (GBM), also called glioblastoma, is the most common cancer malignancy with neuroectodermal origin, showing a considerable variability in age of onset, severity, histological features, and ability to metastasizes [1]. Currently, temozolomide (TMZ) is used as the gold standard chemotherapeutic for the management of GBM among other drugs, i.e., bevacizumab and carmustine [2]. Unfortunately, the characteristics of GBM significantly contribute to treatment failure and poor outcomes for GBM patients [3]. In addition, TMZ has adverse side effects due to the high toxicity and development of resistance over time [2,3]. To improve the effect of TMZ, many studies have combined it with radiation, other chemotherapeutic drugs, and phytochemicals, but many patients still develop drug resistance [1,4]. Moreover, the distinctive tumor microenvironment of GBM stimulates an intrastromal and intratumoral hypoxia cycle, which impairs drug delivery to tumor cells [5,6] and involves aberrant dysregulation of cellular signal transduction pathways [6,7].

Upon oxidative stress, cells protect themselves via a sophisticated intracellular antioxidant system that involves the regulation of glutathione (GSH)/glutathione peroxidase4 (GPX4) [6,8]. Cancer cells exhibit increased levels of intracellular reactive oxygen species (ROS) due to their hypermetabolism [9,10], leading to high expression of aquaporins, NADPH oxidases (NOX) and GPX4 compared to normal cells [11,12,13]. Cancer cells use GSH as a substrate to reduce oxidation products and to suppress cell death, which leads to resistance against therapy [11,12]. An excess of ROS can damage biomembranes and propagate lipid peroxidation chain reactions, and eventually oxidize GSH to Glutathione disulfide (GSSG) [7,14]. To maintain cellular redox homeostasis, GSSG is subsequently reduced to GSH by GPX4 and NADPH/H+, which balances the levels of oxidative mediators [15]. The reduction and activation of antioxidant-related signaling molecules, such as GSH/GPX4 and nuclear factor erythroid 2-related factor 2 (Nrf2), plays an important role in TMZ resistance, as it can turn the oxidative-mediated response on and off, dependent on the intracellular redox status [16,17,18,19]. High intracellular GSH/GPX4 levels in GBM cells lead to epithelial–mesenchymal transition, which results in tumor progression, metastasis and chemoresistance [20]. A possible solution to TMZ toxicity and resistance could be the combination of TMZ with other drugs or therapies that could modulate the GSH/GPX4 levels [21].

Besides chemotherapeutic drugs, an exogenous source of reactive oxygen and nitrogen species (RONS), such as cold atmospheric plasma (CAP) treatment, in combination with TMZ has been found to inhibit cell growth and induce cell cycle arrest in human GBM cell lines LN18, LN229 and U87-MG in vitro [22]. Moreover, the combination of both treatments can reduce cell migration and increase the expression of surface integrins αvβ3 and αvβ5 in GBM cell lines [23].

Recently, our group demonstrated the effect of CAP treatment on three-dimensional (3D) human glioblastoma spheroid models, and we found that plasma-generated short-lived and long-lived species, such as ^•^OH, O_2_^•^^−^, NO_2_^−^, NO_3_^−^ and H_2_O_2_, inhibited spheroid growth and reduced cell migration [24]. Other studies have shown the effect of plasma-treated liquids (PTLs); more particularly, plasma-activated phosphate-buffered saline (P-A PBS), sodium chloride 0.9% (P-A NaCl) and culture medium (PTM) on human colorectal (HCT-116) and ovarian cancer (SKOV-3) spheroid models. These studies show that PTL treatment induced cell death and reduction in spheroid growth [25,26,27]. However, despite the great interest of these works, none of these studies provided precise information on the plasma-induced cell death mechanisms in a spheroid model, or the role of the antioxidant machinery in human GBM tumor models. Indeed, inhibiting the antioxidant machinery, such as GSH/GPX4, plays an important role to enhance the efficacy of TMZ. Hence, there is no evidence whether CAP can sensitize GBM cells towards TMZ, by inhibiting the antioxidant machinery. Indeed, the only two papers published in the literature on the combination of CAP with TMZ [22,23], only used 2D cell cultures, did not focus on the cell death mechanisms and the role of the antioxidant machinery, and did not reveal whether CAP can restore the sensitivity of TMZ-resistant GBM in 3D tumor-like tissues.

In the present study, we therefore focus on the combined CAP and TMZ treatment of both TMZ-sensitive and TMZ-resistant GBM spheroids, which provide a more complex tumor microenvironment than cell monolayers. We determined the role of plasma-generated RONS in the induction of oxidative stress-mediated cell death in GBM cells by studying the depletion of intracellular GSH/GPX4 and corresponding DNA damage. Hence, this study provides evidence of the molecular anticancer mechanisms in combined CAP–TMZ treatment and improves our general understanding of the effect of CAP in cancer therapy.

## 2. Materials and Methods

### 2.1. Cell Lines

We obtained the human glioblastoma (GBM) cell lines (U251, LN18, LN229, U87-MG and T98G) from the Cell Line Service GmbH. All cells were cultured in Dulbecco’s Modified Eagle Medium (DMEM) supplemented with 10% Fetal bovine serum (FBS), 1% glutamine, 1% penicillin (100 IU/mL) and streptomycin (100 mg/mL), all from Gibco™ (ThermoFisher Scientific, Waltham, MA, USA). Cell cultures were maintained in a 5% CO_2_ environment at 37 °C, with 95% relative humidity. After thawing the cells, these were passaged 2–3 times to reach their regular growth rate prior to experiments. The cells U251 (passage 55 + 21), LN18 (passage 6), LN229 (passage 20), U87-MG (passage 32 + 9) and T98G (passage 15) were grown as monolayer cultures, allowed to adhere and maintained to approximately 90% confluency. In addition, cells are regularly tested for the absence of mycoplasma contamination using the MycoAlert detection kit (Lonza, Basel, Switzerland). In general, the cells were kept in culture for 4–5 weeks and, after that, a new vial of cells was thawed.

Cells were transduced with the Nuclight Red Lentivirus reagent (Essen Biosciences, Ann Arbor, MI, USA) using their standard transduction protocol.

### 2.2. Indirect Plasma Treatment and TMZ Treatment of 2D GBM Cell Lines

We treated phosphate buffered saline (PBS) using the kINPen^®^ IND plasma jet (INP Greifswald/neoplas tools GmbH, Greifswald, Germany), as described previously [28,29]. Briefly, plasma was generated with argon gas at a flow rate of 3 LPM (liters per minute) with 10 mm distance between the nozzle of the plasma jet device and the liquid surface. A 12-well round adherent bottom plate was used to expose 2 mL PBS to the plasma jet. The plasma treatment time was 9 min for all treatments. The cells were grown until they reached confluency and 2 × 10^5^ cells/mL were seeded into a flat bottom 96-well plate. The following day, we replaced the culture medium with different percentages of plasma-treated PBS (PT-PBS), i.e., 200 μL cell culture media containing 20, 10, 5, 2.5 and 1.2% phosphate buffer saline (PBS), or with the standard GBM treatment chemotherapy drug temozolomide (TMZ) (Sigma-Aldrich, Missouri, USA). TMZ was resuspended at 20 mg/mL in 100% DMSO, aliquoted and stored at −20 °C at a concentration of 100 mM. In our previous studies [28,30,31] we quantified the level of H_2_O_2_ and NO_2_^-^ in PT-PBS in the same conditions (i.e., 9 min treatment with the kINPen plasma jet), and we detected H_2_O_2_ and NO_2_^−^ concentrations of 430 μM and 50 μM, respectively, inside the PT-PBS.

### 2.3. IC_50_ Estimation

We determined the cell viability on the monolayers of all five GBM cell lines by the 3-[4,5- dimethylthiazol-2yl]-2,5-diphenyltetrazolium bromide (MTT) assay (Sigma-Aldrich, St. Louis, MO, USA), according to the manufacturer’s instructions. All five GBM cell lines were stimulated for up to 24 h with different percentages of PT-PBS and different concentrations of TMZ (as described above) to determine the half maximal inhibitory concentration (IC_50_) value for each treatment. Subsequently, we combined the optimized doses of the single treatments (PT-PBS and TMZ), i.e., the cells are exposed to PT-PBS and TMZ simultaneously.

### 2.4. Combination Index

To determine the synergistic cytotoxic effect of the combination of TMZ and PT-PBS in all five GBM cell lines, the combination index (CI) value was calculated based on [32]. In general, CI values < 0.1 indicate very strong synergism, CI = 0.1–0.3 strong synergism, CI = 0.3–0.7 synergism, CI = 0.7–0.9 slight synergism and CI = 0.9–1.1 nearly additive, while CI = 1.1–1.45 refers to slight to moderate antagonism [32,33].

### 2.5. Spheroid Formation

We used a non-adherent technique for the formation of spheroids [24]. Briefly, cell suspensions were prepared in culture medium supplemented with 0.24% methyl-cellulose in DMEM to enhance spheroid formation. Methyl-cellulose was prepared as previously described [24,34]. An amount of 100 µL of cell suspension (5000 cells/well) were seeded in ultra-low attachment (ULA) 96-well plates (round bottom, Corning Costar, Corning, One Riverfront Plaza, Corning, NY, USA) and centrifuged for 10 min at 1000 rpm. The plates were incubated at 37 ℃ in a 5% CO_2_ humidified atmosphere and spheroids were formed for 3 days.

### 2.6. Indirect and Direct Plasma Treatments, in Combination with TMZ, of 3D Spheroids

After determining the optimal doses, we investigated the effect of these doses on the 3D spheroid model. Before treatment, three-day-old spheroids were washed twice with PBS to remove the culture medium. For the indirect treatment, PBS was first treated with the kINPen plasma jet and then transferred to the spheroids. More specifically, 200 µL of PT-PBS was applied and the plate was placed in the incubator (37 ℃, 5% CO_2_) for 1 h. After this, fresh warm complete culture medium containing TMZ was added to each well. In our 2D cell cultures, we replaced the culture medium with different percentages of plasma-treated PBS (PT-PBS), i.e., 200 μL cell culture medium containing 20, 10, 5, 2.5 and 1.2% phosphate buffer saline (PBS). The cells were stimulated for up to 24 h because they start to die after 4–5 h of treatment and up to 24 h almost 80 to 90% of the cells were dead in the PT-PBS + TMZ combination. We followed a protocol based on our previous work (Shaw et al. 2019) [29]. In case of indirect treatment (by the kINPen) of the 3D spheroids, the PT-PBS together with TMZ have no effect on the reduction in tumor size. Therefore, we incubated the spheroids first with PT-PBS for 1 h and then we added TMZ together with fresh medium. Additionally, in that case, our experiments revealed that indirect treatment by the kINPen plasma jet was not strong enough to induce significant reduction in the tumor size or cell death in the spheroids. Moreover, longer treatment times could increase the stress in cells due to lack of nutrients and growth factors, which could bias the results on cell death assessed. To monitor cell viability in spheroids for 7 days it is important to replace the PBS with culture medium immediately after the treatment. Thus, in the next experiments, the spheroids were subjected to direct plasma treatment. We found that direct treatment by the kINPen plasma jet disturbed the spheroid stability due to the high flow rate of argon gas during plasma generation. Thus, we applied a softer plasma jet device, i.e., the COST plasma jet to the wells containing the spheroids in 200 µL PBS [24,35,36]. Briefly, the plasma was sustained at 250 root mean square voltage (V_RMS_) and an operating frequency of 13.56 MHz. It was operated with a feed gas of 1 LPM He with 5% H_2_O vapor admixture achieved by passing part of the He through an H_2_O-filled Drechsel flask. As reported previously by our group, the administration of H_2_O vapor helps to generate a higher amount of RONS [24]. The spheroids, contained in 200 µL of PBS in a ULA plate, were exposed to a single CAP treatment with the COST jet for 3 min, or to TMZ alone at 40 µM for U251 and 72 µM for U87-MG and T98G (optimized dosage from the IC_50_ value as described below) or the combination of both (CAP for 3 min + TMZ, 40 µM for U251 and 72 µM for U87-MG and T98G). Spheroids in 200 µL of untreated PBS were used as negative controls. We kept untreated spheroids in untreated PBS (“vehicle” solution) for the same period of time as the spheroids treated with plasma/TMZ. The PBS used in this case was not treated with plasma, serving as a negative “vehicle” control for our experiments. After the direct plasma treatment, spheroids remained in the treated solution and were incubated for 60 min. For the direct treatment with the COST plasma jet, spheroids in 200 µL of PBS per well were directly treated with plasma, and the plate was placed in the incubator (37 °C, 5% CO2) for 1 h by following our standard protocol [24]. After this, fresh medium containing the corresponding concentration of TMZ was added to the wells for the assessment of combinational treatments. The incubation time was chosen based on the time it takes to treat a whole plate per cell line with all treatment conditions (around 50–60 min). By the end of treatment, the first treated spheroids had already spent about 1 h in the plasma-treated solution. Longer times than 1 h could increase the stress in cells, due to lack of nutrients and growth factors, which could affect the spheroid growth. The RONS present in the COST jet have been previously described [24]. This study revealed that H_2_O_2_, NO_2_^−^ and NO_3_^−^ are the main RONS present in the PT-PBS and are responsible for the biological effects, and their concentrations for the same conditions as used in this paper were measured to be 1200 μM, 5 μM, and 6 μM, respectively. This is different than for the PT-PBS treated by the kINPen (see above) and can explain why the PT-PBS produced by the COST jet is more effective for the spheroids.

### 2.7. Cytotoxicity Assay

We used the Cytotox Green reagent (50 nM, Essen Biosciences) in media for real-time quantification of cell death. This cell-permeable compound dye binds to nuclear and mitochondrial DNA and becomes strongly fluorescent upon oxidation. The plates were incubated in the IncuCyte Live-Cell Analysis System (Sartorius, Ann Arbor, Michigan, MI, USA) and spheroid growth was followed during seven days after treatment. Data were analyzed with the IncuCyte ZOOM version 2016B (Essen BioScience) as well as ImageJ software to collect the following metrics: (1) spheroid core area, corresponding to the proliferative region only (red live cells; calculated by drawing a circle at the edge of the bright red core of the spheroid using ImageJ); (2) total spheroid area, corresponding to the combined viable, proliferative spheroid core and the Cytotox Green^+^ region (total spheroid region measured from phase contrast images); and (3) the amount of Cytotox Green^+^ in treated spheroids (confluence percentage of the image area occupied by green objects using ImageJ software) normalized to the untreated control at each time point, representing the cytotoxic effect of the treatment.

### 2.8. Estimation of Intracellular Reactive Oxygen Species (ROS) Levels

To measure the release of intracellular ROS in the spheroids, we used CellROX Green Reagent (Thermo Fisher Scientific, Waltham, MA, USA) in media. CellROX is a DNA-binding cell-permeant dye and exhibits bright green photostable fluorescence upon oxidation by reactive oxygen species. We followed the incorporation of the fluorescent probe by the spheroids over a period of 24 h, following CAP, TMZ alone and the combined treatment, with or without the presence of the ROS inhibitor, with a preincubation of 1 h for N-Acetyl-cysteine (NAC, 2.5 µM, Sigma-Aldrich (St. Louis, MO, USA)). A total of 60 min after treatment, we added the culture medium containing 2.5 µM CellROX Green reagent. We analyzed the green fluorescence with the IncuCyte ZOOM (Editor 2018A) software.

### 2.9. Assessment of Glutathione Levels and 8-Hydroxy-2′-Deoxyguanosine Levels

A total of 24 h post treatments, we washed the spheroids twice with PBS and incubated them in 50 µL TrypLE express reagent (12604-021, Life Technologies Europe B.V., Bleiswijk, The Netherlands) for 15 min at 37 °C in a 96-well plate. Immediately after, we added fresh warm culture medium and we dissociated the spheroids with gentle mechanical force using a Pasteur pipette. After the cells were centrifuged at 1000 rpm up to 3 min, we measured the glutathione (GSH) and 8-hydroxy-2′-deoxyguanosine (8-OHdG) levels. We quantified the total GSH and genomic DNA in 10^4^ cells from dissociated spheroids. We measured the GSH level using a luminescence-based assay (Promega, Madison, WI, USA) following a standard molecular biology protocol [37,38]. We used the genomic DNA extracted from the same number of tumor cells for the detection of the 8-OHdG level using an oxidative DNA damage ELISA kit (Cell Biolabs, inc. San Diego, CA, USA).

### 2.10. Immunohistochemical Analysis for Ki-67

We collected the spheroids 24 h after treatment and fixed them with 4% paraformaldehyde for 24 h at 4 °C. We transferred the fixed spheroids to a 4% agarose mold as described before [24]. The agarose pads were embedded in paraffin and cut into 5-μm sections. For staining with the cell proliferation marker Ki-67, antigen retrieval was performed with citrate buffer (10 mM, pH 6), at 96 °C for 20 min. Sections were permeabilized in 0.1% Tween-20 and blocked with EnVision FLEX peroxidase blocking reagent (Dako, SM842, Produktionsvej, Glostrup, Denmark) for 10 min at room temperature (RT). The slides were then incubated with 1:75 dilution of the mouse Anti-Human Ki-67 Antigen (Clone MIB-1, Agilent, Santa Clara, CA, USA) for 40 min at RT. The samples were then incubated with the secondary antibody Envision Flex HRP, Agilent (Santa Clara, CA, USA) for 30 min at RT. Chromogen (substrate) DAB (9511, CINtec histology, Heidelberg, Germany) was added to reveal the reaction. Haematoxylin was used to counter-stain. The sections were imaged with a Zeiss AxioImager Z1 microscope (Carl Zeiss, Göttingen, Germany) equipped with an AxioCam MR ver.3.0. The number of ki67^+^ and haematoxylin^+^ cells was counted using ImageJ software. %ki67^+^ cells = (number ki67^+^ cells/number haematoxylin^+^ cells)*100%.

### 2.11. Immunohistochemistry of GPX4 Enzyme and 8-Oxo-2′-Deoxyguanosine

We evaluated the expression of GXP4 and 8-Oxo-2′-deoxyguanosine (8-oxo-dG) in sections from the spheroid tumors. The slides were first incubated with blocking buffer, i.e., 5% bovine Serum Albumin (BSA) in PBS for 8-oxo-dG (incubated for 2 h) and EnVision FLEX peroxidase blocking reagent (Dako, SM842) for GPX4 (incubated for 10 min) at RT, followed by an incubation with the rabbit- mAb anti-Glutathione peroxidase 4 (1:200; ab125066, Abcam, Cambridge, UK) and anti-8-oxo-dG (1:50; 206461 Abcam), overnight at 4 °C. Subsequently, the stained slides were washed with 0.05% TritonX-100 for 8-oxo-dG and 0.1% Tween-20 for GPX4. The slides for 8-oxo-dG staining were incubated with Alexa Fluor 488 secondary antibody (1:500; ab150113, Abcam) at RT for 1 h and the slides with GPX4 staining were incubated with the secondary antibody Envision Flex HRP, Agilent (Santa Clara, CA, USA) for 30 min at RT. All sections were subsequently imaged with a Zeiss AxioImager Z1 microscope (Carl Zeiss, Göttingen, Germany) equipped with an AxioCam MR ver.3.0 (Carl Zeiss, Göttingen, Germany). The mean percentage area was counted using ImageJ (National Institutes of Health, Public Domain, BSD-2) software.

### 2.12. Statistical Analyses

All experiments were performed in at least three independent biological replicates and are shown as the mean ± standard error of the mean (SEM). Statistical analysis was performed by one-way analysis of variance (ANOVA) with Tukey’s comparison analysis. The data were considered statistically significant when ∗ = *p* ≤ 0.05; ∗∗ = *p* ≤ 0.01; ∗∗∗ = *p* ≤ 0.001. *p*-values for individual treatment are compared to untreated controls.

## 3. Results

### 3.1. PT-PBS Effectively Enhances the Activity of TMZ in 2D Cell Cultures

In this study, we used PBS as plasma-treated liquid (PTL) because PBS prevents cells rupturing or shriveling up due to osmosis. The anticancer effects of other PTL such as NaCl or plasma-treated media have been studied as well [39,40,41,42]. However, plasma can change the acidity of NaCl solutions, decreasing the stability of soluble RONS [39]. We believe PT-PBS might be more suitable for practical applications in a clinical setting than (commonly used) plasma-treated media or NaCl [25], because of its higher stability of RONS [30]. Indeed, PBS and other phosphate buffers are used to generate PTL due to their buffering properties that control the acidification during or after plasma exposure [40,41]. Moreover, the components present in PBS enhance the membrane electropermeabilization [43] by decreasing the phospholipid free energy barrier, which favors the transfer of PT-PBS radicals into the cell and cause oxidative stress that mediates cell death. For instance, Griseti et al. has shown that PT-PBS induced a fast-occurring and more pronounced cell death, visible within deeper layers of the 3D spheroid models [25].

We used the U251, LN18, and LN229 U87-MG and T98G cell lines. We analyzed the cytotoxic effect of PT-PBS alone, TMZ alone and their combination, on all five GBM monolayer cell cultures with the MTT assay. A total of 24 h incubation with PT-PBS decreased the cell viability in all GBM cell lines in a dose-dependent manner, in a concentration range of 1.2–20% PT-PBS in 200 μL culture medium (Figure 1a). The same applies to TMZ treatment in a concentration range of 6–100 μM added to cells in 200 μL culture medium (Figure 1b).

U87-MG and T98G cells were less sensitive to TMZ than the other three cell lines (Figure 1b). Upon comparing both dose responses, the U251, LN18, and LN229 cells revealed similar sensitivity to PT-PBS or TMZ treatment, with IC_50_ values of 12% PT-PBS or 45 μM TMZ treatment (Figure 1c,d). The U87-MG and T98G cells treated with PT-PBS or TMZ presented slightly higher IC_50_ values of 18% PT-PBS for U87-MG and 24% for T98G, and approx. 80 μM TMZ for both cell lines (Figure 1c,d). As controls, we used 10% non-treated PBS in the culture media for all GBM cells. We observed slight cell death in the case of 20% untreated PBS, but in 10% untreated PBS there was no cell death observed, as shown in Appendix A. The DMSO concentration in all the TMZ, plasma, and combined plasma + TMZ treatments was maintained at 0.25% for all experiments. These results show that U251, LN18 and LN229 are sensitive to both treatments. In contrast, U87-MG is resistant only to TMZ, while T98G is resistant to TMZ, as well as to PT-PBS at lower doses. Additionally, the dose–response cytotoxicity curve, as well as the IC50 values of TMZ and PT-PBS monotherapy (Figure 1a–d), showed that U251 (IC50 = 38.1 μM), LN18 (IC50 = 45 μM), and LN229 (IC50 = 39 μM) were more sensitive to low concentrations of TMZ. In contrast, U87-MG (IC50 = 78 μM) and T98G (IC50 = 80 μM) required higher concentrations of TMZ to evoke the same cytotoxic response. This is in good agreement with the literature [4,44]. Based on these results, we have categorized these five GBM cell lines into two groups: TMZ-resistant (IC50 value >50 μM) and TMZ-sensitive (IC50 value <50 μM). From here on, we will use these terms to refer to these cell lines in this study.

After determining the IC_50_ values for PT-PBS and TMZ, we further compared the cytotoxicity of single or combination treatments of PT-PBS with TMZ. We selected a slightly lower concentration of PT-PBS than its IC_50_ values (i.e., we fixed it at 10% for the TMZ-sensitive cell lines; 15% for U87-MG and 20% for the T98G cell line) and we varied the concentrations of TMZ in a range lower than its obtained IC_50_ values (i.e., 40, 20 and 10 µM for the TMZ-sensitive cell lines, and 72, 36 and 18 µM for the TMZ-resistant cell lines). As shown in Figure 1e,g, TMZ (40 µM) and 10% PT-PBS treatment alone had a cytotoxic effect in U251, LN18, and LN229 (i.e., the TMZ-sensitive cell lines) after 24 h (approx. 35 to 40% cell death, close to the IC_50_ values). Interestingly, the combination of PT-PBS and TMZ showed synergistic toxicity, inducing up to 85–90% (*p* ≤ 0.001) cell death in all TMZ-sensitive cell lines. Similarly, in case of U87-MG and T98G, TMZ 72 µM and 15% or 20% PT-PBS treatments alone were cytotoxic after 24 h (approx. 25 to 35% cell death). The combination treatment of PT-PBS and TMZ again showed synergistic toxicity, resulting in up to 78% (*p* ≤ 0.001) cell death in U87-MG and 62% (*p* ≤ 0.01) cell death in T98G (Figure 1h,i). Appendix A presents the calculated CI, based on [32]. For the lowest TMZ concentrations, the CI is around one for LN18, U87-MG and T98G, indicating an additive effect, while it is around 0.6–0.7 for U251 and LN229, representing synergism. The CI drops for most cell lines upon higher TMZ concentrations, down to 0.3, indicating a strong synergism. Altogether, these results suggest the synergistic cytotoxic action of PT-PBS and TMZ in both TMZ-sensitive and TMZ-resistant cell lines.

### 3.2. Indirect Plasma Treatment (PT-PBS) and TMZ Treatment do Not Cause Significant Damage or Cell Death in 3D Spheroids

We tested the cytotoxicity of PT-PBS (10%) and TMZ (40 µM) alone and in combination on U251 (TMZ-sensitive) spheroids, using the doses obtained from the 2D experiments. The experimental design of the indirect plasma treatment of the spheroids is illustrated in Figure 2a. The total spheroid area was measured to assess changes in spheroid size due to cell death and destruction of the spheroid architecture (Figure 2b–e).

The individual treatment of U251 spheroids with 10% PT-PBS or 40 µM TMZ did not induce significant cytotoxic effects (*p* = ns) and did not inhibit spheroid growth (Figure 2b–d). We only observed a slight increase in cell death for the combined treatment (*p* = ns, Figure 2e).

Large tumor spheroids with a diameter greater than 500 μm normally consist of three layers, i.e., proliferating and quiescent regions and a necrotic core. Proliferating cells provide the driving force for tumor growth. It is a target of interest for tumor studies, since a lot of activities occur in this region. Quiescent cells have no growth or active motion but still consume nutrients. The necrotic core is comprised of dead cells which are regarded only as viscoelastic material without living activities. When the nutrient environment changes, proliferating cells may become quiescent cells and eventually die due to the limited distribution of oxygen and nutrients. In addition, quiescent cells may convert to the proliferating type if sufficient nutrients return, which leads to heterogeneous situations within the spheroid core. This heterogeneous nature of viable cells within a spheroid is also observed in our study (Figure 2b and Figure 3e). A bright red signal (representative of viable cells) is observed at the edge of the spheroid, and it decreases or disappears towards the necrotic center. After mono or combinational treatments, we observed cell debris around the spheroids, which are mostly formed by dead cells (positive for the cell death stain Cytotox Green). We defined the area under the sharp and tight edges of the spheroid as the “spheroid core area”, whereas the “total spheroid area” is the sum of both “spheroid core area” and the area of debris. Hence, these treatments alone or in combination were not enough to inhibit spheroid growth or decrease the spheroid core area. The reason is that the treated spheroids receive only long-lived reactive species, produced upon interaction between plasma and the treated liquid. Indeed, it is reported that 3D spheroids are more resistant to treatment than cells in monolayers. Direct treatment, however, delivers both short-lived and long-lived species, as well as ultraviolet radiation, charged particles, and electromagnetic fields, with higher chances to induce the desired response in spheroids. Thus, for further treatments, we directly applied the COST plasma jet on U251 (TMZ-sensitive) and U87-MG and T98G (TMZ-resistant) spheroid models. We selected these three cell lines because of their ability to form tight spheroids.

### 3.3. Direct Plasma Treatment Is Cytotoxic for Both TMZ-Sensitive and TMZ-Resistant GBM Spheroids

Here, we evaluate the effect of CAP in combination with TMZ on U251 (TMZ-sensitive) and U87-MG and T98G (TMZ-resistant) spheroids, which offer a closer architecture to the natural tumor microenvironment than 2D cell cultures. The experimental setup is schematically illustrated in Figure 3a. Treating the spheroids with TMZ (40 µM for U251 and 72 µM for U87-MG and T98G) alone has virtually no effects, with only a slight reduction in the total spheroid area and core area and a small fraction of dead cells in all three cell lines (Figure 3b–e), evidencing a low response to TMZ treatment. On the other hand, 3-min CAP treatment shows a clear reduction in total spheroid area and core area, and a larger fraction of dead cells, in U251 and U87-MG, but not in T98G, suggesting resistance of the T98G spheroids to CAP treatment.

However, when 3 min CAP and TMZ (40 µM for U251, and 72 µM for U87-MG and T98G) treatments were combined, significant cytotoxicity (expressed as the reduction in total spheroid and core area) was achieved in both TMZ-sensitive (U251, reduction in total spheroid (*p* ≤ 0.01, core area *p* ≤ 0.001) and TMZ-resistant spheroid models (U87-MG, reduction in total spheroid (*p* ≤ 0.01), core area (*p* ≤ 0.001) and T98G, reduction in total spheroid (*p* ≤ 0.001), core area (*p* ≤ 0.01)) (Figure 3b,c). Similarly, both the TMZ-sensitive and TMZ-resistant spheroids presented elevated intensity of Cytotox green (U251 (*p* ≤ 0.001), U87-MG (*p* ≤ 0.001) and T98G (*p* ≤ 0.01)) (Figure 3d) in the combination treatment, which is also quite clear from the fluorescence microscopy analysis (Figure 3e). These findings indicate that the combined treatment induces cell death within the spheroids.

### 3.4. Ki-67 Expression Is Reduced in TMZ-Sensitive and TMZ-Resistant GBM Spheroids

To understand the specific inhibitory therapeutic effects of CAP and TMZ treatment, we assessed the expression of the proliferation marker Ki-67 in the spheroids by immunohistochemistry. We observed a reduced percentage of Ki-67^+^ cells after the individual 3-min CAP (*p* ≤ 0.01 for U251, and *p* ≤ 0.05 for U87-MG and T98G) and TMZ (*p* ≤ 0.05 for U251, and non-significant for U87-MG and T98G) treatments compared with the untreated controls, but the effect was much more pronounced in all three spheroids after the combined treatment (*p* ≤ 0.001) (Figure 4).

### 3.5. Induction of Intracellular ROS Causes GSH/GPX4 Inhibition in TMZ-Sensitive and TMZ-Resistant GBM Spheroids

To evaluate the change in the intracellular antioxidant machinery and the release of intracellular ROS upon treatment, we quantified the intracellular ROS levels, the depletion of glutathione levels (GSH) and the suppression of GPX4 expression upon treatment.

We measured the intracellular ROS levels using CellROX^®^ Green reagent (Thermo Fisher Scientific, Walthamcity, MA, USA). This cell-permeable compound dye binds to nuclear and mitochondrial DNA and becomes strongly fluorescent upon oxidation. In case of intracellular ROS release, the combination treatment 3 min CAP + TMZ shows higher fluorescence intensity when compared to the CAP and TMZ treatment alone for both TMZ-sensitive (U251, *p* ≤ 0.001) and TMZ-resistant (U87-MG, *p* ≤ 0.001; and T98G, *p* ≤ 0.001) spheroids (Figure 5). This indicates that CAP synergistically enhances the therapeutic effect of TMZ against GBM spheroids. The ROS scavenger NAC reversed the effects of the combination therapy in both TMZ-sensitive and TMZ-resistant spheroids.

We observed that the combination therapy significantly (for U251 cells *p* ≤ 0.01, for U87-MG cells *p* ≤ 0.01 and for T98G cells *p* ≤ 0.05) decreases the GSH levels (Figure 6a) and the protein expression of GPX4 (Figure 6b,c) in both TMZ-sensitive (U251, *p* ≤ 0.01) and TMZ-resistant (U87-MG, *p* ≤ 0.001 and T98G, *p* ≤ 0.01) spheroids.

The effects of combined treatment seem to be restored by the addition of NAC inhibitor, indicating that CAP synergistically enhances the therapeutic effect of TMZ in three different GBM spheroids.

### 3.6. Combination of CAP and TMZ Causes DNA Oxidation in TMZ-Sensitive and TMZ-Resistant GBM Spheroids

To gain further insight into the impairment of the antioxidant function by CAP-induced DNA damage in TMZ-sensitive and TMZ-resistant GBM spheroids, we measured the levels of 8-hydroxy-2′-deoxyguanosine (8-OHdG) and 8-Oxo-2′-deoxyguanosine (8-oxo-dG), which are the most representative biomarkers of oxidative damage to DNA. Resistance of GBM to TMZ therapy was found to be closely related to the GSH/GPX4 (antioxidant machinery) system of the cell [45]. In our recent paper [31] we showed that CAP triggered cell death in a dose- and time-dependent manner, which was due to increased ROS levels and an inhibition of the GSH/GPX4 antioxidant machinery. Reactive oxygen species accumulation is regarded as one hallmark of oxidative stress-mediated cell death. Many data show that various ROS scavengers can entirely avoid cell death and cellular ROS accumulation [23]. Similarly, administration of TMZ downregulates the antioxidant-related signaling molecules, such as GSH/GPX4, which play an important role in cell death [46,47,48]. Thus, it could be possible that the combination treatment induces excessive intracellular RONS, which may subsequently have a critical role in oxidative stress-induced cell death. Therefore, together with increasing the endogenous ROS environment, inhibiting the antioxidant system results in the accumulation of hydroxyl (OH) radicals and promotes DNA oxidation (see graphical abstract).

The combination treatment of 3-min CAP and TMZ resulted in a higher expression of 8-OHdG when compared to the CAP and TMZ treatments alone (U251 (*p* ≤ 0.001), U87-MG (*p* ≤ 0.001) and T98G (*p* ≤ 0.001), Figure 7a–f), as well as higher levels of 8-oxo-dG formation (U251 (*p* ≤ 0.001), U87-MG (*p* ≤ 0.001) and T98G (*p* ≤ 0.001); Figure 7g–i), in both TMZ-sensitive and TMZ-resistant spheroids. Again, this confirms the role of CAP in amplifying the DNA damage upon TMZ treatment and leads to a strong synergistic effect (CI = 0.2 to 0.4; see Appendix A).

## 4. Discussion

The GBM resistance to TMZ is the most important cause of chemotherapy failure [46,47]. Thus, to determine whether the combined treatment of CAP and TMZ reduced the cell viability of GBM cells, we applied indirect plasma treatment (PT-PBS) in combination with TMZ to the five different GBM cell lines. Two types of plasma treatment are applied in cancer research: direct and indirect treatment. In the case of direct treatment, cells or tissue are directly treated with the plasma source. In indirect treatment, PTLs have shown promising therapeutic efficacy against drug resistant cancers, such as GBM, lung cancer, leukemia, melanoma, and pancreatic adenocarcinomas, and receive growing scientific interest [29,48,49,50]. Most of these studies are carried out on 2D monolayer cell cultures, while only a few studies have considered the biological microenvironment of the tumors, which can be mimicked using the 3D tumor spheroid model [24,25,51,52,53,54]. The tumor microenvironment plays a key role in the response to treatment, regulating tumor progression and metastatic processes [55]. Thus, it is possible that therapies developed using only 2D cell cultures do not meet the requirements to achieve the desired response in in vivo models. A 3D tumor spheroid model can bridge this gap because, in this model, cells are grown as aggregates of single or multiple cell types. This allows 3D cell–cell contact, and proliferation in a more physiological geometry that stimulates the production of extracellular matrix proteins and enhances intercellular communication [56]. When evaluating the biological effects for the purpose of quantifying patient treatment efficacy, 3D spheroid assays may thus better reflect the response of cells within a tumor than 2D cell cultures. Thus far, however, no combinational studies of plasma with TMZ have been performed on GBM spheroid models.

We believe CAP enhances the effect of TMZ because it acts via the generation and local deposition of various ROS on the target, leading to changes in the redox environment and redox-mediated signaling towards tumor cell death. We have previously shown that CAP decreases the phospholipid free energy barrier of cell membranes, affecting the membrane integrity and the overall tumor growth [29]. We have also shown that both short-lived and long-lived species delivered by CAP effectively inhibit tumor growth and reduce cell migration in GBM spheroids [24].

Altogether, these data demonstrate that direct plasma treatment has more cytotoxic effects in 3D GBM spheroids than indirect plasma treatment (PT-PBS). Indeed, during direct plasma treatment, the combination of short-lived and long-lived species, such as ^•^OH, O, O_2_^•−^, O_2_, O_3_, NO^•^, NO_2_^•^, NO_2_^−^, NO_3_^−^ and H_2_O_2_, work simultaneously on the targeted cells and tissue. Moreover, the short-lived reactive species, such as ^•^OH, O, O_2_^•−^, O_2_, and NO^•^, lead to subsequent reaction on the tumor surface, which causes self-perpetuation of toxicity after being in direct contact with CAP [57], while the cytotoxicity effects produced by indirect treatment can only be due to long-lived species, such as H_2_O_2_ and NO_2_^−^/NO_3_^−^.

Interestingly, the addition of NAC restores the effect of the combined treatment to the effect of TMZ only in U251 and U87-MG, and to the level of CAP only in T98G. This might indicate some differences in mechanism in the response between the different cell lines but, to elucidate these differences, more dedicated studies would be needed. In the literature, it was shown that T98G was more resistant to TMZ, caused by lower ROS levels and a higher total antioxidant capacity and GSH concentration [44]. Therefore, increasing exogenous ROS levels together with inhibiting the antioxidant defense system could overcome this therapy resistance. In the case of combined treatment, we found significant inhibition of Ki-67^+^ cells. However, in the presence of NAC, the treatment effects were reduced to the level of CAP treatment only. NAC was not able to fully circumvent the combined effect of the treatment, as its addition reduced the ROS to levels similar to those in CAP only. It is thus possible that other signaling molecules were involved in the response evoked. More studies could help to further elucidate this response.

Cancer cells are metabolically hyperactive, they produce high levels of ROS, and are under intrinsic oxidative stress, which makes them more vulnerable to oxidative stress by exogenous RONS, as produced by CAP [13,58,59]. The combination of CAP and TMZ treatment may increase the intracellular ROS levels, which leads to the depletion of the intracellular GSH level, and GPX4 expression, and consequently to cell death. The enhanced intracellular ROS levels in response to CAP + TMZ treatment can further result in the depletion of GSH levels and the suppression of GPX4 expression, which causes oxidative stress-mediated cell death [48,60]. GPX4 inactivation is one of the key features of cell death, which occurs either through a drop in GSH levels and/or by direct oxidative modification of the GPX4 enzyme.

Furthermore, to see the result of impairment of the antioxidant system on DNA damage in TMZ-sensitive and TMZ-resistant GBM spheroids, we measured the levels of 8-hydroxy-2′-deoxyguanosine (8-OHdG) and 8-Oxo-2′-deoxyguanosine (8-oxo-dG), which are the most representative biomarkers of oxidative damage to DNA. Again, this confirms the role of CAP in amplifying the DNA damage to TMZ treatment and led to a strong synergistic effect.

## 5. Conclusions

Temozolomide (TMZ) is an alkylating agent, used in the treatment of GBM, which induces oxidative stress-mediated cell death. A recurring issue with current chemotherapeutics used for GBM treatment is the adverse side effects induced due to the high toxicity of these drugs, and the developed resistance over time. To address this problem, we investigated whether the combined treatment of cold atmospheric plasma (CAP) and TMZ has potential to overcome TMZ resistance in GBM. To achieve our aim, we applied indirect (PT-PBS) as well as direct plasma treatment in combination with TMZ on both TMZ-sensitive and TMZ-resistant GBM cells. Interestingly, we found that the combination of PT-PBS enhances the efficacy of latter towards both TMZ-sensitive and TMZ-resistant cells in vitro. However, PT-PBS alone or in combination was not effective on both U251 (TMZ-sensitive) and U87-MG and T98G (TMZ-resistant) 3D spheroid models, because 3D spheroids are more resistant to treatment than cells in monolayers. To evaluate the response of plasma on tumor spheroids, we applied direct CAP treatment on both TMZ-sensitive and TMZ-resistant spheroids. The reason is that direct plasma treatment delivers a complex mixture of long-lived and short-lived RONS upon direct contact with the biological sample, while during indirect treatment, the biological sample receives only the long-lived reactive species. Our results demonstrate that combined direct CAP + TMZ treatment significantly improves the response of GBM to TMZ treatment in spheroids, as demonstrated by the reduction in spheroid growth. This improvement is attributed to the inhibition of the GSH/GPX4 antioxidant machinery, which leads to DNA oxidation. Indeed, the combination treatment induces the expression of 8-OHdG and the formation of 8-oxo-dG products, likely due to the accumulation of OH radicals induced by CAP. Such attack on DNA can lead to cell death in both TMZ-sensitive and TMZ-resistant spheroid models. The findings presented in this study provide a novel therapeutic strategy for GBM to enhance the efficacy of TMZ through a combination of increasing exogenous ROS and inhibiting the protective antioxidant system.

## Figures and Tables

**Figure 1 cancers-13-01780-f001:**
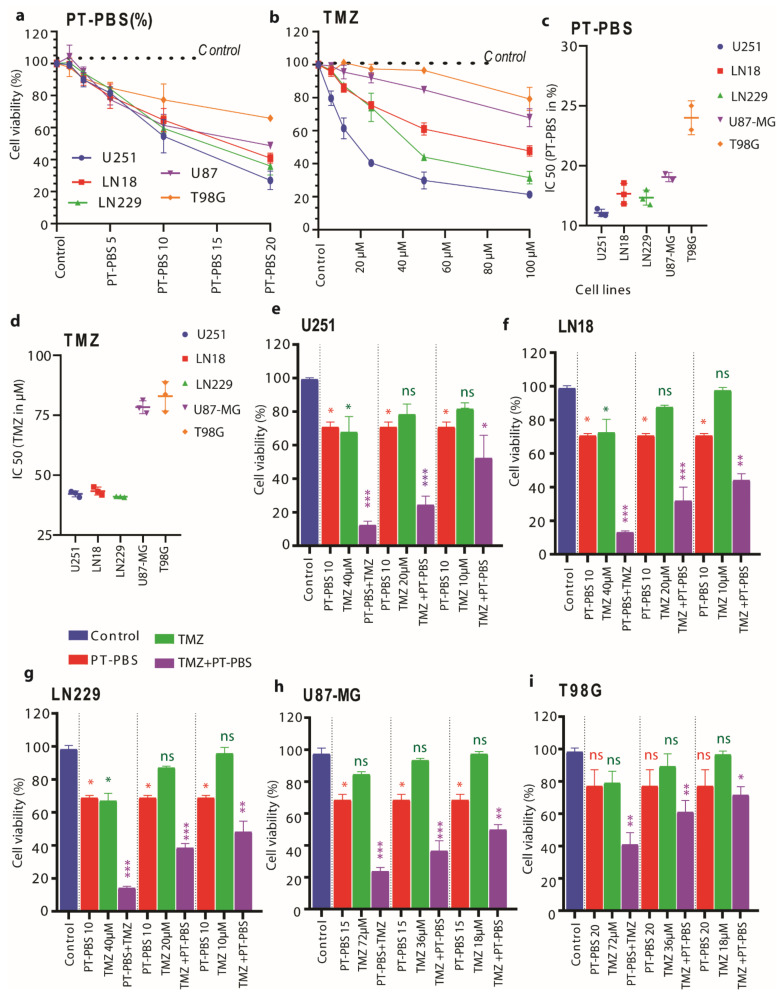
Estimation of optimal doses of TMZ and PT-PBS (kINPen) alone and in combination, for the cytotoxicity of glioblastoma multiforme (GBM) cells in 2D monolayers. We measured the cell cytotoxicity at different doses of (**a**) PT-PBS (the values in the x-axis are in %) and (**b**) TMZ, on U251, LN18 and LN229 (TMZ-sensitive) and U87-MG and T98G (TMZ-resistant) cell lines after 24 h incubation. We also determined the half maximal inhibitory concentration (IC_50_) values of (**c**) PT-PBS and (**d**) TMZ on U251, LN18 and LN229 (TMZ-sensitive) and U87-MG and T98G (TMZ-resistant) cell lines. We analyzed the cell viability at a fixed dose of PT-PBS (10%), and varying doses of TMZ (i.e., 40 µM, 20 µM and 10 µM) alone and in combination, for TMZ-sensitive cells, i.e., (**e**) U251, (**f**) LN18, and (**g**) LN229, as well as for PT-PBS (at 15 or 20%) and varying doses of TMZ (i.e., 72 µM, 36 µM and 18 µM) alone and in combination, for TMZ-resistant cells, i.e., (**h**) U87-MG and (**i**) T98G, after 24 h incubation. The results are derived from three independent biological replicates and are shown as the mean ± standard error of the mean (SEM). Statistical analysis was performed by one-way analysis of variance (ANOVA) with Tukey’s comparison analysis. Statistically significant differences were found between the untreated control and corresponding treated samples. * = *p* ≤ 0.05; ** = *p* ≤ 0.01; *** = *p* ≤ 0.001, ns = not significant. PT-PBS: plasma-treated phosphate buffered saline, TMZ: Temozolomide.

**Figure 2 cancers-13-01780-f002:**
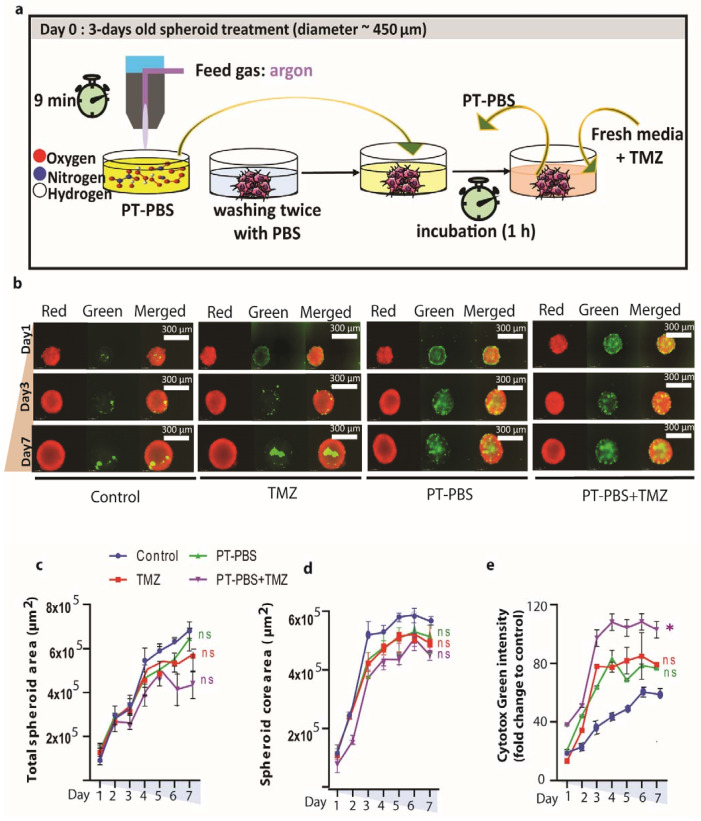
Effect of indirect plasma treatment (PT-PBS, kINPen), TMZ alone and their combination (PT-PBS + TMZ) on U251 human 3D GBM spheroids. (**a**) Schematic presentation of the indirect treatment setup; (**b**) representative images of U251 3D spheroids following indirect plasma treatment: red image of spheroid represents live cells, green represents dead cells within the spheroid, and merged represents the combined image of red and green; (**c**–**e**) quantification of spheroid area: (**c**) total spheroid area of viable cells + dead cells in the spheroid, (**d**) area of the spheroid core (viable cells), and (**e**) Cytotox green intensity in the spheroids (dead cell confluency). Statistical analysis was performed using one-way analysis of variance (ANOVA) with Tukey’s comparison analysis. The results are derived from three independent biological replicates and are shown as the mean ± standard error of the mean (SEM). Statistically significant differences were found between the untreated control and corresponding treated samples. * = *p* ≤ 0.05; ns = not significant.

**Figure 3 cancers-13-01780-f003:**
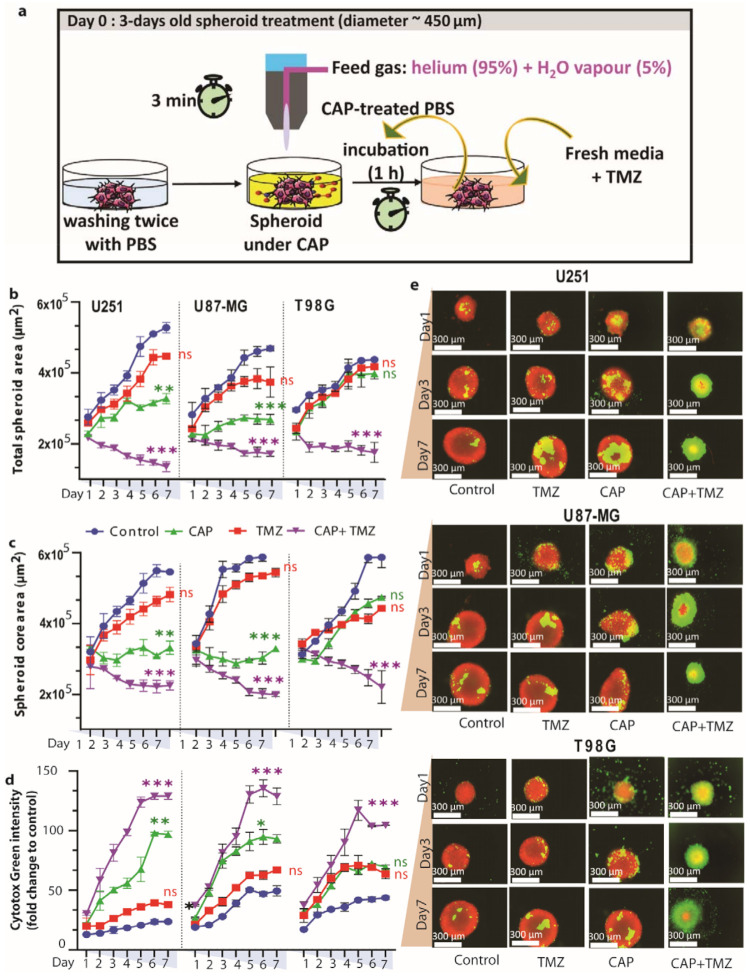
Effect of direct 3 min cold atmospheric plasma (CAP) (COST jet) treatment, TMZ (40 µM for U251, and 72 µM for U87-MG and T98G) and their combination on human TMZ-sensitive (U251) and TMZ-resistant (U87-MG and T98G) GBM spheroids. (**a**) Schematic representation of the CAP treatment setup; (**b**–**d**) quantification of spheroid area: (**b**) total spheroid area, (**c**) area of the spheroid core, and (**d**) Cytotox green intensity (dead cell confluence) in U251, U87-MG and T98G spheroids. Statistical analysis was performed using one-way analysis of variance (ANOVA) with Tukey’s comparison analysis. The results are derived from three independent biological replicates and are shown as the mean ± standard error of the mean (SEM). Statistically significant differences were found between the untreated control and corresponding treated samples. * = *p* ≤ 0.05; ** = *p* ≤ 0.01; *** = *p* ≤ 0.001, ns = not significant. (**e**) Representative images of U251, U87-MG and T98G spheroids following CAP treatment, in flowing convergence image: red represents live cells, green represents dead cells within the spheroid.

**Figure 4 cancers-13-01780-f004:**
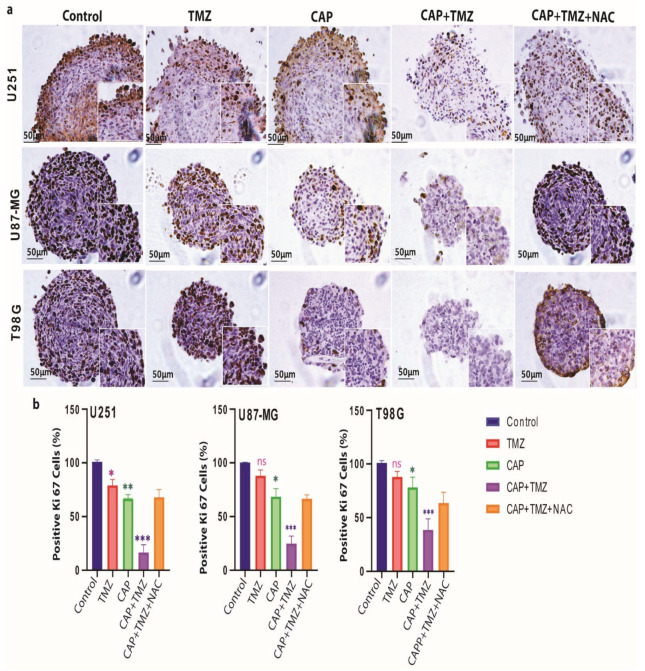
Effect of CAP (COST jet), TMZ (40 µM for U251, and 72 µM for U87-MG and T98G) and their combination treatment on the expression of the proliferative marker Ki-67 in GBM spheroids; (**a**) representative images of Ki-67 staining, showing the reduction in the proliferation marker Ki67^+^ in U251, U87-MG and T98G spheroids compared to control, following TMZ, CAP, CAP + TMZ and CAP + TMZ + NAC treatment. The enlarged image demonstrates variable ki67 staining. (**b**) The staining was scored using ImageJ software. Data are representative of two independent experiments, 4–6 spheroids per condition. Mean ± SD. Statistically significant differences were found between the untreated control and corresponding treated samples. * = *p* ≤ 0.05; ** = *p* ≤ 0.01; *** = *p* ≤ 0.001, ns = not significant. NAC: N-Acetyl-L-cysteine.

**Figure 5 cancers-13-01780-f005:**
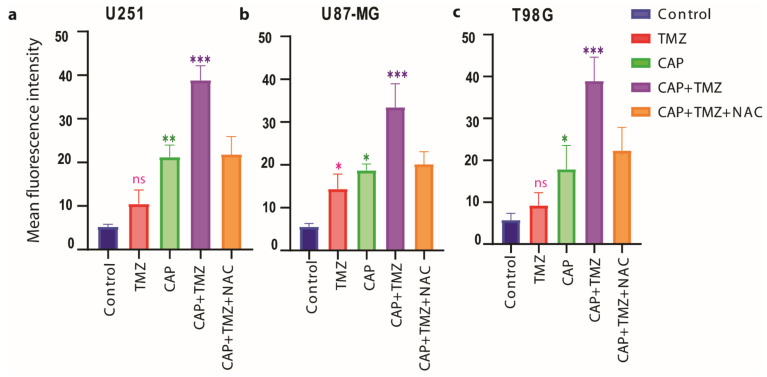
Changes in intracellular reactive oxygen species (ROS) content upon treatment with CAP (3 min, COST jet), TMZ (40 µM for U251, and 72 µM for U87-MG and T98G) alone and their combination (CAP + TMZ), in the presence or absence of NAC, in (**a**) U251 (TMZ-sensitive), (**b**) U87-MG and (**c**) T98G (TMZ-resistant) spheroids. A fluorescent Cell^®^Rox green probe was used to measure the percentage of ROS in treated versus untreated controls. Statistical analysis was performed using one-way analysis of variance (ANOVA), with Tukey’s comparison analysis. Data are representative of three independent experiments; mean ± SD. Statistically significant differences were found between the untreated control and corresponding treated samples. * = *p* ≤ 0.05; ** = *p* ≤ 0.01; *** = *p* ≤ 0.001, ns = not significant.

**Figure 6 cancers-13-01780-f006:**
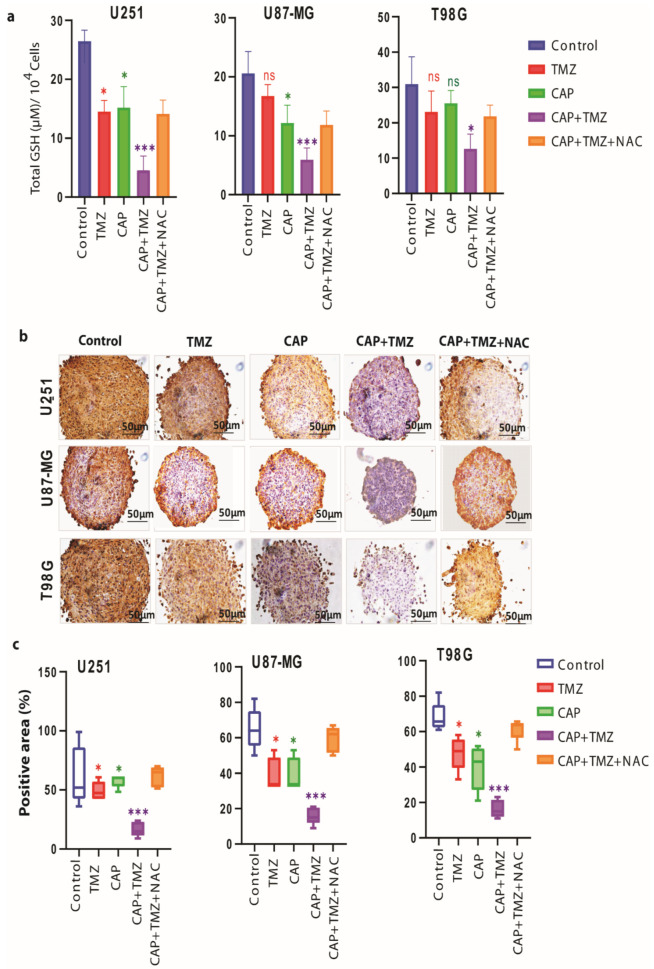
Changes in intracellular GSH levels and cellular GPX4 protein expression upon treatment with CAP (3 min, COST jet), TMZ (40 µM for U251, and 72 µM for U87-MG and T98G) alone and their combination (CAP + TMZ) in the presence or absence of NAC, in U251 (TMZ-sensitive), U87-MG and T98G (TMZ-resistant) spheroids. (**a**) Cellular levels of glutathione (GSH) upon treatment. Statistical analysis was performed using one-way analysis of variance (ANOVA), with Tukey’s comparison analysis. The data were considered significantly different when * = *p* ≤ 0.05; *** = *p* ≤ 0.001. (**b**) Histochemical staining, showing the reduction in the GPX4 protein expression (brown color) in the U251, U87-MG and T98G spheroids compared to control, following TMZ, CAP, CAP + TMZ and CAP + TMZ + NAC treatment. (**c**) GPX4 protein expression quantified using ImageJ software in CAP, TMZ alone and their combination (CAP + TMZ), as well as CAP + TMZ + NAC, versus untreated cells (control). Data are representative of two independent experiments, 4–6 spheroids per condition. Mean ± SD. Statistically significant differences were found between the untreated control and corresponding treated samples; ns = non-significant.

**Figure 7 cancers-13-01780-f007:**
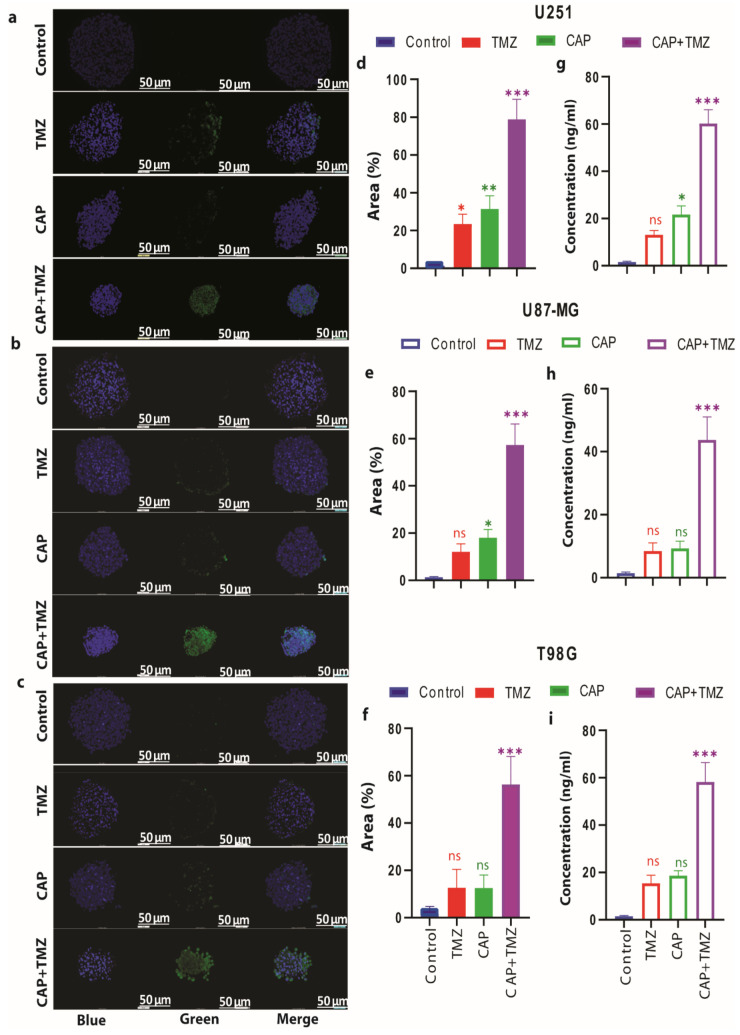
The effect of CAP (3 min, COST jet), TMZ (40 µM for U251, and 72 M for U87-MG and T98G) alone and their combination (CAP + TMZ) on the expression of oxidative stress-mediated DNA damage (DNA oxidation) in U251 (TMZ-sensitive), U87-MG and T98G (TMZ-resistant) spheroids. (**a**–**c**) Representative images of immunohistochemistry staining, showing 8-Oxo-2′-deoxyguanosine (8-oxo-dG) over-expression (DNA oxidation) in U251, U87-MG and T98G spheroids upon TMZ (40 µM) alone and their combination (CAP + TMZ) versus untreated cells (control). 8-oxo-dG expression represents green (oxidized DNA), blue (4′,6-diamidino-2-phenylindole (DAPI)) represents the nuclear counterstain. (**d**–**f**) The staining was scored using ImageJ software (3 min CAP, TMZ (40 µM) alone and their combination (CAP + TMZ) versus untreated cells (control). Data are representative of two independent experiments, 4–6 spheroids per condition. Mean ± SD; * = *p* ≤ 0.05; ** = *p* ≤ 0.01; *** = *p* ≤ 0.001. (**g**–**i**) Quantification of 8-hydroxy-2′-deoxyguanosine (8-OHdG) in U251, U87-MG and T98G spheroids upon TMZ (40 µM) alone and their combination (CAP + TMZ) versus untreated cells (control). Statistical analysis was performed using one-way analysis of variance (ANOVA), with Tukey’s comparison analysis. Data representative of two independent experiments, 4–6 spheroids per condition. Mean ± SD. Statistically significant differences were found between the untreated control and corresponding treated samples; ns = not significant.

## Data Availability

The data presented in this study are available on request from the corresponding author.

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
