# Peer review of "Cold Atmospheric Plasma Increases Temozolomide Sensitivity of Three-Dimensional Glioblastoma Spheroids via Oxidative Stress-Mediated DNA Damage"

_cancers, 2021, doi:10.3390/cancers13081780_

Round 1
Reviewer 1 Report
In the present manuscript “Cold atmospheric plasma increases temozolomide sensitivity of 2 three-dimensional glioblastoma spheroids via oxidative stress-mediated DNA damage”, the authors explore Cold atmospheric plasma (CAP) as a promising combinatory therapy against glioblastoma multiforme. For that, they used 2-D and 3-D in vitro models. This work highlight the potential of this therapeutic approach to treat temozolomide (TMZ)-sensitive and -resistant patients.
- In general, the submitted manuscript is very well structured and with careful writing, without major spelling or grammatical errors.
- Introduction provides important and updated theoretical considerations of the different topics necessary for the all understanding of the paper.
- Methodology provides careful details and is clearly described. Some questions on methodology:
- How were TMZ-resistant cells obtained? Please describe or provide appropriate reference. According to lines 196-197, I supposed TMZ-resistant cells lines were also purchased. Please distinguish TMZ-sensitive and -resistant cell lines in section 2.1
- Please provide details on the passage of cells used and frequent testing to check for the presence of mycoplasma contamination.
- Why did you choose PBS as a PTL to irradiate? We now that PBS components can influence ROS conversion and as consequence, antioxidant response and the molecular anticancer mechanisms involved. In a clinical point of view, it is important to study a PTL whose application in clinics is acceptable and makes sense. With this in mind, NaCl is a more appropriate PTL. Please address this topic in the discussion.
- Lines 102-103: How did you manage final volumes or volumes of media in each condition? I.e. For the condition 20% of PBS, implies to use 40ul of PBS and 160ul of medium. For the condition 1.2% of PBS, implies to use 2.4ul of PBS and 197.6ul of medium. If this was the approach, the nutrients available for cells in each condition is quite different, which can represent a bias in the results analysis. Please clarify.
- Lines 123-124: How was this damage evaluated?
- Line 134: “Spheroids in 200 µL of untreated PBS were used as negative controls.” This implies that negative controls were also irradiated, right? It is important to clarify. If not, how can you ensure that CAP did not affected the spheroid stability and integrity?
- Authors said in line 29 “we used a three-dimensional multicellular human spheroid model for the U251 (TMZ-sensitive) and U87-MG and T98G (TMZ-resistant) cells”. This gives the idea that a 3D model with several types of cells mimicking the microenvironment was done. However, methods don’t describe the use of different kinds of cells (e.g. fibroblasts, etc.). Please clarify and do the proper corrections.
- Results are clearly presented. Discussion presents many findings of other authors corroborating the original results presented in this manuscript. Discussion also makes clear the innovative of this work in compared to what is known so far about CAP’s mechanistic anticancer effects. Results are strictly discussed from the 2D model till the 3D approach, highlighting the clear involvement of oxidative stress through ROS promotion and Antioxidative mechanisms suppression. Some questions on results and discussion:
- To ensure the existence or not of a synergistic effect, it would be important to calculate the combinatory index
- Line 269: based on the literature (e.g. https://doi.org/10.1016/j.ddtec.2017.03.002 ), the known spheroid structure includes a necrotic core. Which is your basis to consider the spheroid core as viable cells?
- Although the statistical differences are well explained in the text, their representation in the graphs of figure 3b), c) and d) is quite confusing. There are lines without * which makes it difficult to interpret.
- Line 308: although these results are quite promising and interesting, how do you explain the absence of statistical differences between CAP and CAP+TMZ treatments? The same for intracellular ROS content (Fig 5).
- Line 323: Figure b) results are represented using Boxplots graphs, so they are not represented as mean+/-SD as stated here. Please correct. Same comment in line 362 and 381.
Overall, this manuscript is worth of publication in Cancers journal with minor revisions and clarifications.
Reviewer 2 Report
The manuscript entitled ‘Cold atmospheric plasma increases temozolomide sensitivity of 2 three-dimensional glioblastoma spheroids via oxidative stress-3 mediated DNA damage’ by Shaw and co-workers focusses on the combinatorial treatment of glioblastoma cell lines with a combination of TMZ and plasma treated PBS and the treatment of tumour spheroids with TMZ in combination with exposure to a plasma jet. The authors further investigated the intracellular ROS levels and the antioxidant responses to elucidate the mechanisms of action.
This study provides some very interesting findings on the synergistic mechanisms of TMZ and indirect/direct plasma and could help to develop combinatorial treatments in future. Importantly the authors use tumour spheroids as a model which is more akin to the real tumour environment than a 2D cell layer. The manuscript is very well-written and results are well-presented though in some figures the statistical significance could be shown more clearly.
The authors clearly show the synergistic effects of PT-PBS and TMZ and CAP and TMZ in both TMZ-sensitive and resistant cells. However, I do have some reservations about the conclusion that CAP treatment can sensitise cells to TMZ based on the results shown and feel this needs further clarification which is the reason I would recommend major revision even if most of my comments are minor. Please find my detailed comments below.
- The authors state that ‘Overall, our findings suggest that the combination of CAP with TMZ is a promising combinational therapy to circumvent TMZ resistance of GBM patients.’
The authors show very interesting data on the effects of CAP+TMZ on both TMZ-sensitive and resistant cells. However, in my opinion this study is missing a key experiment to show that the combination can actually circumvent TMZ resistance. In the present experiments the authors are using higher TMZ concentrations (72uM vs 40 uM) for TMZ-resistant cells that show similar effects (ROS, amount of cell death etc) on these resistant cells as the lower concentrations do on TMZ-sensitive cells. The ‘baseline’ situation is therefore comparable between the cells. However, to really show that CAP can sensitise resistant cells to TMZ I think such experiments would also need to be performed at the low TMZ concentrations used for sensitive cells. In my opinion, only by showing an ability to kill TMZ-resistant cells through CAP+TMZ treatment at low TMZ concentrations, can you demonstrate that the CAP is sensitising the cells to TMZ and circumventing TMZ-resistance. Otherwise I think these conclusions may need to be revised and stated in a more cautious manner.
Introduction
- Line 75: ‘Indeed, inhibiting the antioxidant machinery, such as GSH/GPX4, plays an important role to sensitise TMZ. Hence, there is no evidence whether CAP can sensitise TMZ by inhibiting the antioxidant machinery..’ I think, it is the cells that are sensitised to TMZ not the TMZ that is sensitised and these sentences should be phrased as ‘Indeed, inhibiting the antioxidant machinery, such as GSH/GPX4, plays an important role to sensitise cells to TMZ. Hence, there is no evidence whether CAP can sensitise cells to TMZ by inhibiting the antioxidant machinery..’ or similar.
Experimental procedure
- ‘All five GBM cell lines were stimulated for up to 24 hours with different percentages of PT-PBS and different concentrations of TMZ…’ – did the authors test different stimulation times up to 24h? Why was 1h exposure to PT-PBS selected for the spheroid treatment? Did the authors perform any optimization experiments on the duration of exposure to PT-PBS prior to addition of the TMZ? Would greater damage have been caused to the spheroids if they had been left in the PT-PBS for more than1h?
- Line 110 ‘we combined the optimised doses of the single treatments (PT-PBS and TMZ).’ – were cells exposed to PT-PBS and TMZ simultaneously? In this context line 221 ‘Altogether, these results suggest that PT-PBS effectively sensitises the GBM cells to TMZ, both for TMZ-sensitive and TMZ-resistant cell lines.’ Can the authors really conclude from this that PT-PBS sensitises the cells to TMZ – could it be just a synergistic effect of the 2 treatments?
- Do the authors have any information on the reactive species composition in the PT-PBS? Why did they choose to treat the PBS for 9 min? Did the authors compare PBS treated with the kinPEN and the COST jet? Since the initial experiments using indirect treatment were performed with the kinPEN, it would be very interesting to know if the respective PBS behaved similarly.
Materials and Methods
- 2.6. Cytotoxicity Assay – ‘We used the Cytotox Green reagent (50 nM, Essen Biosciences) for real-time quantification of cell death.’ – please specify how the staining procedure was performed.
- 2.7 ‘To measure the release of intracellular ROS in the spheroids, we used CellROX Green Reagent (Thermo Fisher 148 Scientific, Massachusetts, USA)’ – please specify how the staining procedure was performed
- 2.7 ‘with or without the presence of 2.5 μM of the ROS inhibitor N-Acetyl-L-cysteine (NAC)’ – were cells pre-incubated with NAC or was this added at the same time?
Results and conclusions
- Fig 1 a and b: please consider showing these graphs with a linear concentration scale as x-axis rather than a ‘category’ scale. Plotting against category distorts the curve and makes the decrease in viability appear much steeper as all data points are equidistant on the x-axis while they are actually quite different in concentration.
- Fig 1 statistical analysis: please clarify in the figures and the figure legend which comparison the analysis refers to: comparison to the control, comparison to the other 2 treatments within the same group or comparison to the same treatment in another group. The ns for ‘TMZ only’ treatments in particular seems a bit confusing here – what are those compared to as they seem quite different from controls and ‘PT-PBS only’ and have small error bars. And what about those bars which are marked with neither an asterisk nor an ns?
- Line 257 ‘The reason is that the treated spheroids receive only long-lived reactive species, produced upon interaction between plasma and the treated liquid.’ – as already pointed out above, is it possible that effects would have been stronger if the spheroids had been pre-treated with PT-PBS for longer periods of time or exposed to PT-PBS and TMZ simultaneously as seems to be the case for the 2D cell models? Please explain your rationale for performing the experiment in this way.
- Fig 2 e: The bracket in this figure suggests no significant difference of PT-PBS + TMZ to PT-PBS and TMZ alone. But what about to the control where the difference seems much bigger? In general, please clarify in 2c-e if the ‘ns’ shown refers to PT-PBS+ TMZ compared to CTL as the brackets are not fully clear.
- Line 281 ‘area and core area and a small fraction of death cells in all three cell lines’ = ‘area and core area and a small fraction of dead cells in all three cell lines’
- Fig 3b-d: figure legends to indicate which colour/symbols correspond to which treatment are missing. Please show these in the figure as done in previous figures or explain in the legend below.
- Fig 4: quite interestingly addition of NAC seems to restore the effect of the combined treatment to the effect of TMZ-only in U251 and U87mg but to the level of CAP only in T98G. Could this indicate some differences in mechanism in the response between the different cell lines?
- Line 314 ‘Interestingly, the treatment effects were lost in the presence of the ROS inhibitor NAC, indicating that the ROS induced by CAP are most probably responsible for the reduction of Ki-67+ cells and for enhancing the TMZ sensitivity of the GBM spheroids.’ Does this statement need to be revised when considering the data obtained in Fig. 5 which suggests that the addition of NAC decreases ROS to levels similar to those in CAP only…?
- Line 350 ‘Again, the effects were lost in the presence of the ROS inhibitor NAC, indicating that the excessive RONS induced by CAP lead to inactivation of the GPX4/GSH antioxidant system in GBM, which is most probably responsible for sensitising GBM to TMZ.’ And 366 ‘Indeed, as shown above, CAP-generated RONS cause the failure of GPX4 …’ Similar to above, I’m not sure I would agree with this based on the data: GSH levels are very similar between TMZ only and CAP only. Why do the authors conclude that GPX4/GSH inactivation are the result of excessive RONS induced by CAP when RONS levels between TMZ and CAP appear very similar and effects on GSH/GPX4 seem comparable between the 2?
- Line 372 ‘Again, this confirms the role of CAP in enhancing the cell response to TMZ treatment.’ I would be more cautious and state that this confirms the synergistic effect of CAP and TMZ but in my opinion it is difficult to determine from the current data whether CAP enhances the effect of TMZ or vice versa…
- 6a and 6c: statistics: the brackets are not very clear – do they refer to comparison of CAP+TMZ to CTL as suggested in the legend? CAP+TMZ+NAC is marked as ‘ns’ – does this refer to comparison to the CTL as well? What about the bars that show neither an asterisk nor a ns?
- 7: the images of immunohistochemical staining of the spheroids are very dark and even the DAPI counterstain is difficult to see. Is it possible to provide better images as the current ones provide only very limited information? The brackets in figures d-I seem to indicate comparison of CAP+TMZ to TMZ only but I think according to the legend it should be to the untreated control? Please clarify.
Round 2
Reviewer 2 Report
Thank you. I am satisfied with the responses to the reviewer’s comments and the changes the authors have made to the manuscript, which I think have improved the clarity. The additional analysis of the combination index has added to the quality of the data presented and the authors now avoid referring to ‘sensitization of TMZ-resistant cells’, which I believe is prudent based on their current data. I hope the authors are continuing their research on this topic and may be able to show sensitization in future. In my opinion, only some very minor changes are required prior to publication.
- I think 2 of my queries relating to methodology of Cytotox Green and CellROX Green were misunderstood. I was not referring to the principle of the staining procedure but rather asking for details of the methodology: e.g. what volumes and concentrations of Cytotox Green or CellROX green were added (in medium or buffer?), how long were cells stained for etc
- The section which was added in line 163 is confusing: ‘For the direct treatment of the spheroids with the COST plasma jet, we also incubate the spheroids first with PT-PBS for an hour and then TMZ is added with fresh culture medium. More specifically, 200 µL of PT-PBS was applied per spheroid and the plate was placed in the incubator (37 ◦C, 5% CO2) for 1h by following our standard protocol [24].’ Does this mean the cells were treated with both PT-PBS and CAP? or is this just referring to the PBS remaining on the cells for 1hour after CAP treatment before addition of TMZ with medium (in which case the sentences should be changed)? or is this section referring to indirect not direct treatment?
- For overall clarity it might be beneficial to add to the figure legends whether the treatment was by kINPen or COST jet (e.g. Figure 1: Estimation of optimal doses of TMZ and PT-PBS (kINPen)..; Figure 2. Effect of indirect plasma treatment (PT-PBS, kINPen)…; Figure 3. Effect of direct 3 min CAP treatment (COST jet)…). This is just a suggestion.
- A few spelling or grammatical errors were found in the modified sections and should be corrected:
- 99 In general, the cells were kept in culture for 4-5 weeks and after that, a new vial of cells were thawed = In general, the cells were kept in culture for 4-5 weeks and after that, a new vial of cells was thawed
- 363 but the effect was much more pronounced all three spheroid after the combined treatment = but the effect was much more pronounced in all three spheroid after the combined treatment
- 381 This indicate that the CAP synergistically enhances the therapeutic effect of TMZ against GBM spheroids = This indicates that the CAP synergistically enhances the therapeutic effect of TMZ against GBM spheroids
- 394 The effects of combined treatment seems to be restored by the addition of NAC inhibitor, indicating that CAP synergistically enhances the therapeutic effect of TMZ in three different GBM spheroids. = The effects of combined treatment seem to be restored by the addition of NAC inhibitor, indicating that CAP synergistically enhances the therapeutic effect of TMZ in three different GBM spheroids.
- 410 Resistant of GBM to TMZ therapy are found to be closely related to GSH/GPX4 (antioxidant machinery) system of the cell [45] = Resistance of GBM to TMZ therapy was found to be closely related to GSH/GPX4 (antioxidant machinery) system of the cell
- 418 Therefore, together with increasing endogenous ROS environment and inhibiting antioxidant system result in the accumulation of hydroxyl (⋅OH) radicals and promote DNA oxidation (see graphical abstract). = Therefore, increasing endogenous ROS environment and inhibiting antioxidant system together result in the accumulation of hydroxyl (⋅OH) radicals and promote DNA oxidation (see graphical abstract). Or Therefore, together with increasing endogenous ROS environment inhibiting antioxidant system results in the accumulation of hydroxyl (⋅OH) radicals and promote DNA oxidation (see graphical abstract).
- 473 However, in presences of NAC = However, in presence of NAC
